

# Plastic response of *Medicago sativa* L. root system traits and cold resistance to simulated rainfall events

Zhensong Li, Liqiang Wan, Shuo Li, Xianglin Li, Feng He and Zongyong Tong

Institute of Animal Science, Chinese Academy of Agricultural Sciences, Beijing, China

## ABSTRACT

Climate change (rainfall events and global warming) affects the survival of alfalfa (*Medicago sativa* L.) in winter. Appropriate water management can quickly reduce the mortality of alfalfa during winter. To determine how changes in water affect the cold resistance of alfalfa, we explored the root system traits under different rainfall events and the effects on cold resistance in three alfalfa cultivars. These were exposed to three simulated rainfall events (SRE) × two phases in a randomized complete block design with six replications. The three cultivars were WL168, WL353 and WL440, and the three SRE were irrigation once every second day ($D_2$), every four days ($D_4$) and every eight days ($D_8$). There were two phases: before cold acclimation and after cold acclimation. Our results demonstrated that a period of exposure to low temperature was required for alfalfa to achieve maximum cold resistance. The root system tended toward herringbone branching under $D_8$, compared with $D_2$ and $D_4$, and demonstrated greater root biomass, crown diameter, root volume, average link length and topological index. Nevertheless, $D_8$ had less lateral root length, root surface area, specific root length, root forks and fractal dimensions. Greater root biomass and topological index were beneficial to cold resistance in alfalfa, while more lateral roots and root forks inhibited its ability to survive winter. Alfalfa roots had higher proline, soluble sugar and starch content in $D_8$ than in $D_2$ and $D_4$. In contrast, there was lower malondialdehyde in $D_8$, indicating that alfalfa had better cold resistance following a longer irrigation interval before winter. After examining root biomass, root system traits and physiological indexes we concluded that WL168 exhibited stronger cold resistance. Our results contribute to greater understanding of root and cold stress, consequently providing references for selection of cultivars and field water management to improve cold resistance of alfalfa in the context of changes in rainfall patterns.

Corresponding author
Feng He, hefeng@caas.cn

## INTRODUCTION

Alfalfa (*Medicago sativa* L.) has spread widely because of its productivity and palatability. The global area planted with alfalfa is about $3.2 \times 10^7$ hm$^2$ and is mainly distributed in the United States, Russia and Argentina (*Russelle, 2001*). The area planted with alfalfa

exceeds $4 \times 10^6$ hm$^2$ in China (*He, 2011*), where it is mainly distributed in the high latitudes. There has been frequent crop failure due to winter conditions in recent years, such as the "Easter freeze" of 2007 in the United States (*Augspurger, 2009*) and the frost in northern China from 2012 to 2020 (*Yang et al., 2019*). The failure of alfalfa to overwinter not only causes huge economic losses (*Castonguay et al., 2006*), but also reduces biological nitrogen fixation and increases nitrous oxide emissions from agronomic ecosystems, leading to an increased risk of global warming (*Crews & Peoples, 2004*; *Robertson, Paul & Harwood, 2000*). The IPCC Fifth Assessment Report showed that the average temperature of the world had risen by 0.85 °C in the past 100 years and by 1 °C in China over the past few decades (*Fang et al., 2018*). Two major factors linked to climate change are likely to affect plant winter survival: changes in precipitation and temperatures (*Bélanger et al., 2002*; *Bélanger et al., 2001*). Extreme low temperature is more likely to reduce the survival rate of alfalfa in winter owing to reduced snowfall, and greater temperature fluctuation can make alfalfa break dormancy prematurely, exposing vulnerable buds to subsequent killing frost and causing sustained damage (*Augspurger, 2009*). Modeling of global climate change has predicted that alfalfa death due to reduced snowfall and greater temperature fluctuation will occur more frequently in the future (*Carol, 2013*; *Ji et al., 2017*).

Alfalfa needs to undergo a period of low temperature and a short photoperiod to obtain its freezing tolerance, and this is known as cold acclimation (*Theocharis, Clement & Barka, 2012*; *Trischuk et al., 2014*). Appropriate cultivation measures are also an effective way to improve cold resistance. Water plays an important role in the winter hardiness of alfalfa because freezing injury is mainly caused by cell dehydration (*Xu et al., 2020a*; *2020b*; *Zhang et al., 2015*). Water can not only affect the cold resistance of alfalfa by changing the morphology and spatial distribution of the root system, but also protect cells from low-temperature damage through physiological metabolic pathways (*Castonguay et al., 2006*). In addition to precipitation amount, the impact on root growth also includes precipitation timing and interval. Research has shown that precipitation events have different effects on various plants, and high-frequency light precipitation events were found to have a greater impact on herbaceous plants (*Schwinning & Sala, 2004*; *Schwinning, Starr & Ehleringer, 2003*). The root system is key to the ability of alfalfa to resist low temperature. Plants can regulate root system development in response to dynamic changes in soil moisture (*Comas et al., 2013*; *Li et al., 2020*). Previous studies have documented that below-ground biomass (BGB) (*Larson & Smith, 1963*; *Liu et al., 2015*), root crown (*Larson & Smith, 1963*; *Liu et al., 2015*; *Schwab et al., 1996*), lateral roots (*Liu et al., 2015*; *Smith, 1951*) and root system spatial distribution (*Castonguay et al., 2006*) all affect the cold resistance of alfalfa. According to the cross acclimation theory (*Kong & Henry, 2019a*; *2019b*), low-frequency heavy precipitation will increase the time plants spend in drought conditions, and so improve the cold resistance of alfalfa. This is because drought can enhance the antioxidant capacity and osmotic regulation of plants, including changes in malondialdehyde (MDA) (*Schwab et al., 1996*), proline (Pro) (*Janska et al., 2010*), soluble sugars (SS) (*Trischuk et al., 2014*) and starch content (*Xu et al., 2020a*), and these are closely related to the cold resistance of alfalfa. MDA reflects the degree of
membrane lipid peroxidation in the cell membrane, and its content is directly proportional to the low temperature injury of alfalfa (*Schwab et al., 1996*). Proline improves the cold resistance of plants by regulating osmotic balance and increasing protein solubility (*Janska et al., 2010*). Soluble sugar acts as osmotic regulator, cryoprotectant, and signaling molecule to stabilize the cell membrane and scavenge reactive oxygen species under low temperature (*Trischuk et al., 2014*). Starch can be broken down into soluble sugar to improve the cold resistance of alfalfa (*Xu et al., 2020a*).

Our experiment simulated the effects of different precipitation patterns on alfalfa root traits and cold resistance. The purpose was to: (1) study the response of root morphology and spatial distribution to different precipitation events; (2) clarify the relationship between root traits and cold resistance; (3) explain the effect of different precipitation patterns on cold resistance; (4) clarify whether precipitation patterns have differences in the root traits of alfalfa cultivars. These are of great significance since the water management could prove important for increasing cold resistance.

# MATERIALS & METHODS

## Experiment location and materials

This experiment was conducted in a controlled greenhouse at the Institute of Animal Science, Chinese Academy of Agricultural Sciences (Beijing, China) from May to September 2020, with 25 °C/20 °C (day/night), 14 h/10 h (light/dark) and photosynthetic photon flux density of 350 $\mu mol \cdot m^{-2} \cdot s^{-1}$ at 60–65% relative humidity. Alfalfa seeds were disinfected with sodium hypochlorite (1% NaClO) for 30 min and washed with deionized water five times. We then selected seeds of the same size and germinated them in a Petri dish with 14 h light and 10 h dark at 25 °C. After 72 h we moved three germinant seeds into one polyvinyl chloride (PVC) pipe with an inner diameter of 18 cm and height of 50 cm. A nylon mesh bag was placed in each pipe (to facilitate later sampling) and this was filled with 2.5 kg of sterilized dry mixture with a volume ratio of 4:1 sandy soil and nutrient soil mixture. The nutrient soil mixture is a cultivation medium (composed of peat moss and lime), named TS1, produced by Klasmann–Deilmann. TS1 contains 1.6% total nitrogen, 0.1% P2O5, 0.2% K2O (N: P: K = 14:10:18) and 91% of organic matter, with a conductivity of 0.9 dS/m and pH of 5.8. The water-holding capacity (WHC) of the mixture was measured to be 38.35%. One plant was left in each PVC pipe 1 week after transplanting according to its height (about 15 cm), and cultivation continued for another 2 weeks before subsequent experimentation. The soil moisture content was kept at 60–65% WHC by weighing the pipe every second day. Weeds and pests were removed regularly.

## Experiment designs and treatments

A randomized complete block design was used, with three alfalfa cultivars (C), three simulated rainfall events (SRE) and two phases. The three alfalfa C were WL168, WL353 and WL440 (with a fall dormancy score of 2, 4 and 6, respectively; provided by Beijing Zhengdao Seed Industry Co., Ltd.). These are commonly planted in large areas of northern China and represent the range in fall dormancy scores of alfalfa C grown in the region.

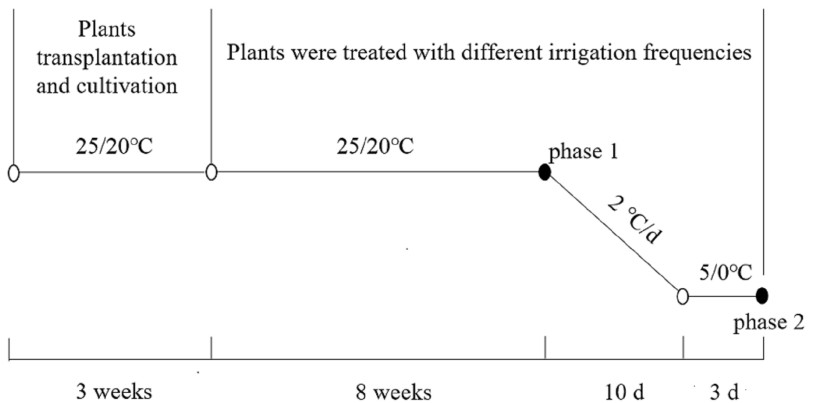

**Figure 1 Schematic diagram of experiment processing and sampling.** 25/20 °C and 5/0 °C represent the temperature of alfalfa during normal growth and cold acclimation, respectivel.

The three SRE were: irrigation once every second day ($D_2$), every four days ($D_4$) and every eight days ($D_8$). $D_2$ represented a high-frequency light precipitation and $D_8$ represented a low-frequency heavy precipitation. The two phases were phase 1 (before cold acclimation) and phase 2 (after cold acclimation). The combined $3 \times 3 \times 2 \times 6$ repetitions = 108 PVC pipes.

According to the preliminary test, the soil moisture content of $D_2$, $D_4$ and $D_8$ before the next irrigation was about 50%, 40% and 30% of WHC respectively. We carried out three irrigation interval treatments, and total irrigation quotas remained the same between different treatments and determined by $D_2$ (keeping the soil moisture content at 60–65% of the WHC by weighing every second day). After 8 weeks we carried out phase 1 sampling ($3 \times 3 \times 6 = 54$ pipes in total, as shown in Fig. 1). The aboveground and underground parts were separated and then carefully removed from the nylon mesh bag in each PVC pipe to minimize damage to the spatial distribution of the root system. The root surface mixture was washed away gently with distilled water by hand and the roots were placed evenly in a transparent acrylic tray with 1,200 mL of distilled water. They were then scanned with a MICROTEK Scan Maker i800plus (Microtek Technology Co., Ltd., Shanghai, China) with a resolution of 600 dpi. Immediately after scanning, about five cm of the root crown was used to determine electrical conductivity and physiological indicators. We divided the sample into two parts, one part was used for the determination of $LT_{50}$, the other part was used for the determination of physiological indicators. The remainder of the root was measured to calculate biomass. The aboveground and underground parts were weighed after being placed in an oven at 65 °C for 48 h and the dry weight were above-ground biomass (AGB) and below-ground biomass (BGB).

The rest half of the experimental plants were moved to an LRH-200-GD low-temperature light incubator (Taihong Medical Instruments, Guangdong, China) for the low-temperature experiment (phase 2). The initial temperature was 25 °C/20 °C (day/night) with a photoperiod of 10 h light and 14 h dark, and the photosynthetic photon flux density was 350 μmol·m$^{-2}$·s$^{-1}$. The temperature was decreased to 5 °C/0 °C
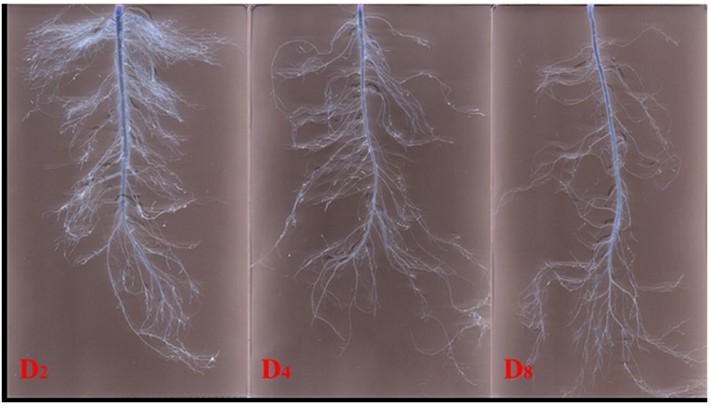

**Figure 2 Scanning image of root system under three simulated rainfall events.** $D_2$, $D_4$ and $D_8$ represent irrigation once every second day, every 4 days and every 8 days respectively.

(day/night) at a rate of 2 °C·d$^{-1}$ and the light intensity was decreased to 150 µmol·m$^{-2}$·s$^{-1}$ at a rate of 20 µmol·m$^{-2}$·s$^{-1}$·d$^{-1}$, simulating the cold adaptation environment of alfalfa. Sampling was carried out after a further 72 h of cold acclimation. During this phase of the experiment, plants were watered as in the previous phase. The root was carefully rinsed by hand with distilled water after the test and the root crown sample was divided into two parts. One part was used for the measurement of electrical conductivity and the other for physiological indicators (kept at −80 °C).

## MEASUREMENTS

### Root morphological indicators

We used a Win-RHIZO 2017a (Regent Instruments, Inc., Quebec, QC, Canada) to analyze the scanned images (Fig. 2). The process included thresholding, framing, editing breakpoints and eliminating loops to obtain the root morphological indicators, root length, root surface area (RSA), root volume (RV), root forks (RF) and average link length (ALL). Topological index (TI) and fractal dimensions (FD) were obtained by calculation.

TI is used to reflect the spatial structure of different root systems and is defined as log altitude (A)/log magnitude (M) where A is the number of links in the longest path from an exterior link to the most basal link of the root system and M is the total number of the exterior links (*Bouma et al., 2001*). When TI is close to 0.5, the root system tends to a dichotomous branching and when close to 1 it tends to herringbone branching. FD were obtained according to the box-dimension method (*Bouma et al., 2001*; *Fitter, 1986*; *Fitter et al., 1991*; *Harrar & Hamami, 2007*).

### Semi-lethal temperature

The semi-lethal temperature (LT$_{50}$, the temperature at which the relative permeability of intracellular ions attains 50%) was used to represent the cold resistance of alfalfa in this study (*Anwer et al., 2016*). We took the five cm underground taproot as the root crown, and then we cut the root crown into nine pieces of 2–3 mm (based on length instead of diameter), and these pieces were put into nine 2-mL centrifuge tubes (*Xu et al.,*

*2020a*). Tubes were placed at 8 °C for 2 h. The subsequent freezing test was conducted in a ZX-5C constant-temperature circulator (Zhixin Instrument, Shanghai, China) under a decreasing series of nine temperatures, and tubes with samples were kept in alcohol for 1.5 h at each temperature. The temperatures of the alcohol were different due to the difference between the $LT_{50}$ under the two treatments. For the samples collected in phase 1, nine temperatures were set to 8 °C, 6 °C, 4 °C, 2 °C, 0 °C, −2 °C, −4 °C, −6 °C and −8 °C. For the samples collected in phase 2, nine temperatures were set to 0 °C, −2 °C, −4 °C, −6 °C, −8 °C, −10 °C, −12 °C, −14 °C and −16 °C. After 1.5 h at the first temperature in each phase, one tube was transferred to storage at that temperature; after 1.5 h at the second temperature another tube was removed for storage at that temperature; and so on until all nine tubes in each phase were stored at their designated temperatures. We then removed the pieces of root crown from each 2-mL tube and placed them in one 15-mL tube and added 5 mL deionized water. This tube was shaken on an HZQ-A gyratory platform shaker (Hengrui Instrument and Equipment, Changzhou, China) at 120 rpm for 12 h at 25 °C. Next used a conductivity meter FE38 (Mettler, Shanghai, China) to measure the electrical conductivity as $EL_1$. The sample was autoclaved at 121 °C for 30 min and, on remeasuring, its electrical conductivity was found to be $EL_2$. The electrical conductivity of deionized water was EL. Relative electrolyte leakage can be calculated according to Eq. (1) and the semi-lethal temperature can be calculated by logistic Eq. (2). In Eq. (2), x is the freezing temperature, y is the relative electrical leakage and A, B and k are constants:

$$\text{Relative electrolyte leakage (\%)} = (EL_1 - EL)/(EL_2 - EL) \times 100 \tag{1}$$

$$y = A/(1 + B \times e^{-kx}) \times 100\% \tag{2}$$

## Root physiological indexes

Physiological indexes were also measured on root crown of alfalfa. Ground the other part of the crown sample kept at −80 °C into powder and then determined their SS (*Buysse & Merckx, 1993*), starch (*Buysse & Merckx, 1993*), MDA (*Dhindsa, Plumb-Dhindsa & Thorpe, 1981*) and Pro content (*Bates, 1973*).

### Malondialdehyde (MDA)

A 0.2-g sample was placed in a 10-ml test tube for determination of MDA. (1) Added 5 mL 0.1% cold trichloroacetic acid (TCA) and then the homogenate was centrifuged at 10,000 r/min for 5 min at 4 °C. (2) To 1 mL aliquot of the supernatant 4 mL 20% TCA containing 0.5% TBA were added. The mixture was heated at 95 °C for 20 min and then quickly cooled in an ice-bath. After centrifuging at 4,000 r/min for 15 min, then 2 mL supernatant was removed to a cuvette and 2 mL deionized water used as control. (3) The absorbance of the supernatant was measured at 450 nm ($OD_{450}$), 532 nm ($OD_{532}$) and 600 nm ($OD_{600}$). The MDA content was measured according to Eq. (3), where

$V_1$ (mL) is the total volume of the supernatant, $V_2$ (mL) is the volume of the measurement and DW (g) is the weight of the freeze-dried sample:

$$MDA \ (nmol/gDW) = [6.452 \times (OD_{532}-OD_{600})-0.559 \times OD_{450}] \times V_1/(V_2 \times DW) \qquad (3)$$

### Proline (Pro)

A 0.2-g sample was placed in a 10-ml test tube for Pro determination. (1) Added 5 mL of 3% aqueous sulfosalicylic acid solution and then transferred the sample to a boiling water bath for 20 min and obtained the Pro extraction after cooling. (2) A 2-mL extraction was moved to another test tube, 2 mL of glacial acetic acid and 2 mL 2.5% of acidic ninhydrin solution were added and the extraction was then transferred to a boiling water bath for 60 min. (3) Added 4 mL of methylbenzene and the tube was shaken after cooling before being centrifuged at 5,000 r/min for 5 min. (4) Measured the absorbance of the supernatant at 520 nm and calculated the Pro content according to the standard curve and Eq. (4), where C is the Pro content obtained from the standard curve, $V_1$ (mL) is the total volume of the extraction, A (mL) is the volume of the measurement and DW (g) is the weight of the freeze-dried sample:

$$Pro \ (\mu g/gDW) = (C \times V_1/A)/DW \qquad (4)$$

### Soluble sugar (SS)

A 0.2-g sample was placed in a 50-mL test tube for determination of the soluble sugar. (1) Added 20 mL deionized water and then transferred the tube to a boiling water bath for 20 min. (2) The tube was centrifuged at 3,500 r/min for 10 min after cooling and the supernatant transferred to a volumetric flask and diluted to 100 mL as an extraction solution (the residue was used later to determine starch content). (3) Placed 1 mL of the extraction solution in another test tube and added 4 mL 2% anthrone ethyl acetate, then placed the tube in a water bath at 90 °C for 15 min. (4) After the tube had cooled and measured the absorbance of the extraction solution at 625 nm. The content of soluble sugar was calculated according to the standard curve and Eq. (5), where C ($\mu$g) is the soluble sugar content according to the standard curve, $V_1$ (mL) is the total volume of the extraction, A (mL) is the volume of the measurement and DW (g) is the weight of the freeze-dried sample:

$$Soluble \ sugar \ (\%) = (C \times V_1)/(A \times DW \times 10^6) \times 100 \qquad (5)$$

### Starch

The remaining residue was moved to a 20-mL test tube for determination of the starch. (1) Added 8 mL of hydrochloric acid and boiled the tube in a water bath for 45 min, then transferred the contents into a volumetric flask and added 8 mL sodium hydroxide and diluted it to 50 mL. (2) Placed 1 mL of the supernatant in a volumetric flask and diluted it to 25 mL as an extraction solution. (3) Added 4 mL of anthrone to 1 mL extraction solution and placed the tube in a boiling water bath for 5 min. (4) After cooling, the absorbance was measured at 625 nm. Starch content was calculated according to the

**Table 1 Above-ground biomass, below-ground biomass and the ratio of below-ground biomass to above-ground biomass among simulated rainfall events or cultivars in phase 1.**

| Treatments | | AGB (g) | BGB (g) | R/S |
|---|---|---|---|---|
| SRE | $D_2$ | 1.60 ± 0.26 | 0.64 ± 0.25[b] | 0.45 ± 0.14[b] |
| | $D_4$ | 1.74 ± 0.34 | 0.87 ± 0.30[a] | 0.51 ± 0.14[b] |
| | $D_8$ | 1.58 ± 0.23 | 1.01 ± 0.27[a] | 0.68 ± 0.17[a] |
| | $p$ value | ns | <0.01 | <0.01 |
| C | WL168 | 1.68 ± 0.19 | 0.95 ± 0.32[a] | 0.57 ± 0.21 |
| | WL353 | 1.76 ± 0.24 | 0.93 ± 0.23[a] | 0.56 ± 0.12 |
| | WL440 | 1.47 ± 0.35 | 0.64 ± 0.20[b] | 0.48 ± 0.16 |
| | $p$ value | ns | <0.01 | ns |

Note:
Mean values ($n$ = 18) ± standard errors of the mean are shown. Different letters represent a significant difference under various SRE and C; ns indicates the difference is not significant; and $p < 0.05$ and $p < 0.01$ indicate significant difference at 0.05 and 0.01 levels, respectively. AGB, above-ground biomass; BGB, below-ground biomass; R/S, the ratio of BGB to AGB; SRE, simulated rainfall events; C, cultivars.

standard curve and Eq. (6), where C (μg) is the starch content according to the standard curve, V (mL) is the total volume of the extraction solution, A (mL) is the volume of the measurement and DW (g) is the weight of the freeze-dried sample:

$$\text{Starch (\%)} = C \times V \times 0.9/(A \times DW \times 10^6) \times 100 \qquad (6)$$

## STATISTICAL ANALYSES

Shapiro–Wilk test and Levene test showed that all data in this experiment obeyed a normal distribution and satisfied the homogeneity of variance. Data in this study were subjected to a two-way analysis of variance between treatments using SPSS 20.0 (SPSS Inc., Chicago, IL, USA). Multiple range tests were performed using least significant differences, and differences were considered significant at $p < 0.05$ and $p < 0.01$; ns was not significant. Principal component analysis (PCA) was also conducted in this experiment.

## RESULTS

### Biomass

After 8 weeks, BGB and the ratio of BGB to AGB (R/S) growth differed significantly ($p < 0.01$) among SRE; nevertheless, there was no significance ($p > 0.05$) in AGB (Table 1). As rainfall intervals increased, BGB and R/S showed an increasing trend. $D_8$ had the highest BGB and R/S (1.01 g and 0.68, respectively) and these were significantly higher than those of $D_2$. There was no major impact on AGB and R/S ($p > 0.05$) of the three C within the same water treatment (Table 1). The BGB of WL440 was 0.64 g, which was significantly ($p < 0.01$) lower than that of WL168 and WL353. Further analysis showed significant interactions between SRE and C on R/S, and the former were found to have played a more important role (Table 2).

### Root morphology

SRE had a little effect ($p > 0.05$) on the primary root length (PRL) of alfalfa (Table 3). $D_2$ had the longest lateral root length (LRL, 247 cm) followed by $D_4$ and $D_8$. The crown

**Table 2 Interaction and simple effect analysis of simulated rainfall events and cultivars on biomass, root traits and $LT_{50}$ among in phase 1.**

|  | SRE×C | SRE | | C | |
|---|---|---|---|---|---|
|  | Sig. | Sig. | PES | Sig. | PES |
| ABG | ns | – | – | – | – |
| BGB | ns | – | – | – | – |
| R/S | <0.05 | <0.01 | 0.250 | 0.113 | 0.085 |
| PRL | ns | – | – | – | – |
| LRL | ns | – | – | – | – |
| CD | ns | | – | – | – |
| RSA | <0.05 | <0.01 | 0.452 | 0.074 | 0.101 |
| RV | ns | – | – | – | – |
| SRL | <0.01 | <0.01 | 0.414 | <0.01 | 0.349 |
| RF | ns | – | – | – | – |
| ALL | ns | – | – | – | – |
| TI | ns | – | – | – | – |
| FD | ns | – | – | – | – |
| $LT_{50}$-phase1 | ns | – | – | – | – |
| $LT_{50}$-phase2 | ns | – | – | – | – |

**Note:**
Significance (Sig.) and partial eta squared (PES) are shown, and ns indicates the difference is not significant. A dash (–) indicates that there was no significant interaction between simulated rainfall events and cultivars. PRL, primary root length; LRL, lateral root length; CD, crown diameter; RSA, root surface area; RV, root volume; SRL, specific root length; RF, root forks; ALL, average link length; TI, topological index; FD, fractal dimensions.

**Table 3 Root morphological traits among simulated rainfall events or cultivars in phase 1.**

| Treatments | | PRL (cm) | LRL (cm) | CD (mm) | RSA (cm²) | RV (cm³) | SRL (cm/g) |
|---|---|---|---|---|---|---|---|
| SRE | $D_2$ | 42.40 ± 2.5 | 247 ± 15[a] | 3.76 ± 0.75[b] | 66.3 ± 9.9[a] | 0.74 ± 0.06[b] | 452 ± 109[a] |
| | $D_4$ | 43.46 ± 2.9 | 214 ± 11[b] | 4.21 ± 0.59[a] | 57.3 ± 9.5[b] | 0.81 ± 0.07[a] | 335 ± 67[b] |
| | $D_8$ | 43.28 ± 3.6 | 162 ± 11[c] | 4.27 ± 0.56[a] | 46.0 ± 9.5[c] | 0.81 ± 0.04[a] | 221 ± 48[c] |
| | *p* value | ns | <0.01 | <0.05 | <0.01 | <0.01 | <0.01 |
| C | WL168 | 42.49 ±2.6 | 185 ± 17[b] | 3.88 ± 0.76 | 57.8 ± 3.6 | 0.80 ± 0.07[a] | 277 ± 69[b] |
| | WL353 | 42.98 ± 2.9 | 214 ± 18[a] | 4.24 ± 0.69 | 59.5 ± 5.0 | 0.80 ± 0.05[a] | 278 ± 49[b] |
| | WL440 | 43.68 ± 3.4 | 225 ± 24[a] | 4.12 ± 0.51 | 52.3 ± 5.9 | 0.75 ± 0.08[b] | 453 ± 114[a] |
| | *p* value | ns | <0.01 | ns | ns | <0.01 | <0.01 |

**Note:**
Mean values (*n* = 18) ± standard errors of the mean are shown. Different letters represent a significant difference under various SRE and C; ns indicates the difference is not significant; and *p* < 0.05 and *p* < 0.01 indicate significant differences at 0.05 and 0.01 levels, respectively.

diameters (CD) of $D_4$ and $D_8$ were 4.21 and 4.27 mm, respectively, significantly ($p < 0.05$) wider than that of $D_2$. RSA varied significantly ($p < 0.01$) among the three SRE; $D_8$ had the minimum at 46.0 cm². $D_4$ and $D_8$ had the same RV, 0.81 cm³, significantly ($p < 0.01$) larger than that of $D_2$. Specific root length (SRL) was significantly ($p < 0.01$) inversely proportional to rainfall events. $D_2$ had the greatest SRL, 452 cm/g, followed by $D_4$ (335 cm/g) and $D_8$ (221 cm/g). The three alfalfa C differed significantly ($p < 0.01$) in LRL,

**Table 4 Root spatial traits between simulated rainfall events or cultivars in phase 1.**

| Treatments | | RF | ALL (cm) | TI | FD |
|---|---|---|---|---|---|
| SRE | $D_2$ | $1{,}572 \pm 242^a$ | $0.119 \pm 0.015^b$ | $0.600 \pm 0.020^b$ | $1.469 \pm 0.043^a$ |
| | $D_4$ | $1{,}325 \pm 275^b$ | $0.117 \pm 0.020^b$ | $0.621 \pm 0.016^a$ | $1.444 \pm 0.034^a$ |
| | $D_8$ | $854 \pm 118^c$ | $0.136 \pm 0.018^a$ | $0.634 \pm 0.022^a$ | $1.390 \pm 0.042^b$ |
| | $p$ value | <0.01 | <0.05 | <0.01 | <0.01 |
| C | WL168 | $1{,}171 \pm 273^b$ | $0.107 \pm 0.021^b$ | $0.622 \pm 0.023$ | $1.437 \pm 0.050$ |
| | WL353 | $1{,}390 \pm 468^a$ | $0.117 \pm 0.028^b$ | $0.610 \pm 0.023$ | $1.447 \pm 0.045$ |
| | WL440 | $1{,}191 \pm 303^b$ | $0.147 \pm 0.020^a$ | $0.623 \pm 0.023$ | $1.419 \pm 0.044$ |
| | $p$ value | <0.01 | <0.01 | ns | ns |

Note:
Mean values ($n = 18$) ± standard errors of the mean are shown. Different letters represent a significant difference under various SRE and C; ns indicates the difference is not significant; and $p < 0.05$ and $p < 0.01$ indicate significant difference at the level of 0.05 and 0.01, respectively.

but not in PRL (Table 3). WL168 showed a significantly ($p < 0.01$) shorter LRL than WL440 and WL353. There was no evidence of a significant difference in CD and RSA between various C. The RV of WL168 and WL353 were both $0.80$ cm$^3$, a value significantly ($p < 0.01$) greater than that of WL440. The SRL of WL168 and WL353 were 277 and 278 cm/g, respectively, much lower than that of WL440. In every treatment combination (SRE × C), the effect of rainfall events on root surface area and specific root length was greater than that of C (Table 2).

## Root system architecture

Basic information about the root system architecture in phase 1 is shown in Table 4. SRE had major effects ($p < 0.01$) on RF, notably on $D_2$ at 1,572. The ALL of $D_8$ was 0.136 cm, which was significantly longer than that of $D_2$ and $D_4$. Across the three SRE, the TI ranged from 0.600 to 0.634 and FD ranged from 1.390 to 1.469. The TI and FD showed a reverse trend. A significant ($p < 0.01$) effect was detected for RF and ALL in various alfalfa C, but there was no significant ($p > 0.05$) difference in TI and FD. WL353 had the maximum RF, 1,390, which was significantly more than RF in WL168 and WL440. In contrast, the ALL of WL353 was 0.117 cm, which was shorter than that of WL440. There were no major interactions between SRE and C in these four indicators (Table 4).

## Semi-lethal temperature

We observed that the cold resistance of alfalfa that had not undergone cold acclimation was relatively weak (Fig. 3). From phase 1 to phase 2 the $LT_{50}$ of three SRE decreased by an average of 8.4 °C, and $D_2$ had the largest decline at 6.1 °C. $D_8$ had the lowest $LT_{50}$ in both phases (0.32 and −6.5 °C, respectively). Simultaneously, there were also significant ($p < 0.01$) differences in the $LT_{50}$ in the two phases among the three alfalfa C. The $LT_{50}$ of WL168 decreased the most and reached 6.9 °C, followed by WL440 and WL353 (5.9 °C and 5.8 °C, respectively). WL168 had the greatest cold resistance among the three C in phase 2, and its $LT_{50}$ was −6.8 °C.

Principal component analysis (PCA, Kaiser–Meyer-Olkin (KMO) value was 0.728 and $p < 0.01$) of 10 variables was used to identify the correlations between the variables and

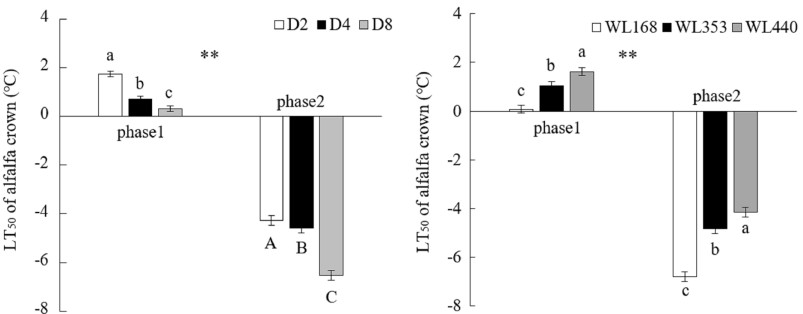

**Figure 3 Semi-lethal temperatures of alfalfa cultivars crowns under different simulated rainfall events.** Mean values ($n$ = 18) ± standard errors of the mean are shown. Different capital letters and lowercase letters indicate a significant difference between three simulated rainfall events and cultivars at the same phase at 0.01 and 0.05 level respectively; asterisks (**) indicate that the same treatment had a significant difference in the two phases at 0.01 level.

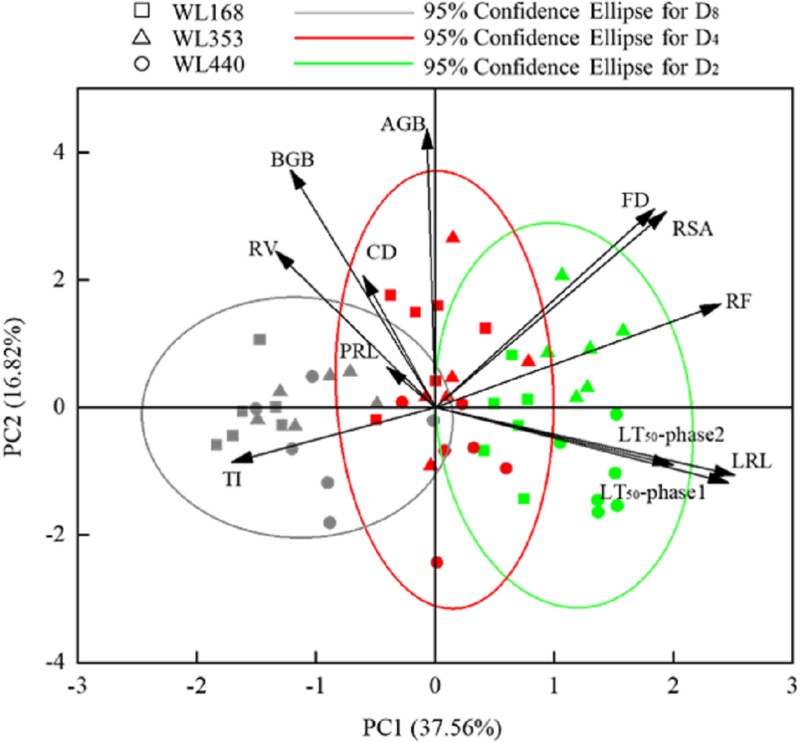

**Figure 4 Principal component analysis of 10 variables and the correlations among variables and LT$_{50}$.** Each arrow represents the eigenvector corresponding to an individual variable. Different colors denote different SRE and shapes refer to cultivars. AGB, above-ground biomass; BGB, below-ground biomass; PRL, primary root length; LRL, lateral root length; CD, crown diameter; RSA, root surface area; RV, root volume; RF, root forks; TI, topological index; FD, fractal dimensions.

LT$_{50}$, which were associated with the first two principal components (Fig. 4). Different colors denote different SRE and shapes refer to cultivars, and SREs play a more important role than cultivars in the difference of root traits. PCA axis 1 primarily reflected the morphological and spatial characteristics of root systems (LRL, RF and TI), which

**Table 5 Component matrix of the first two principal components.**

|  | Component | |
| --- | --- | --- |
|  | 1 | 2 |
| AB | 0.057 | 0.818 |
| BB | −0.403 | 0.537 |
| PRL | −0.121 | 0.195 |
| LRL | 0.826 | −0.288 |
| CD | −0.199 | 0.348 |
| RSA | 0.779 | 0.375 |
| RV | −0.420 | 0.544 |
| RF | 0.910 | 0.095 |
| TI | −0.624 | −0.099 |
| FD | 0.746 | 0.387 |

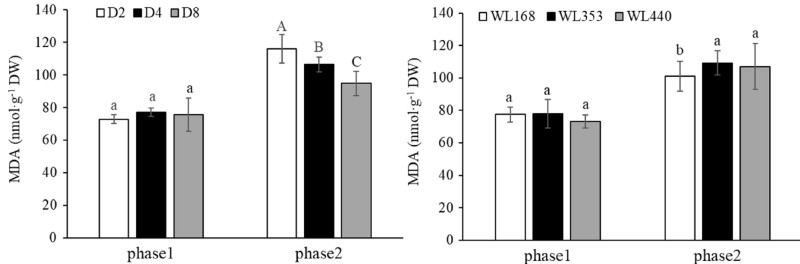

**Figure 5 Malondialdehyde content under various simulated rainfall events for the three cultivars in the two phases.** Mean values ($n = 18$) ± standard errors of the mean are shown. Different capital letters and lowercase letters indicate a significant difference between three simulated rainfall events and cultivars at the same phase at 0.01 and 0.05 level respectively.

accounted for 37.56% of the overall variance in the standardized variables. Axis 2 mainly reflected the biomass of the alfalfa, explaining 16.82% of the standardized variance (Table 5). BGB, PRL and RSA were positively correlated with CD, RV and FD, respectively. LRL was negatively correlated with BGB, CD and RV; and TI was negatively correlated with RSA, RF and FD. $LT_{50}$ was positively correlated with LRL, RSA, FD and RF, but inversely correlated with BGB, CD, RV and TI. Lateral root length, root forks and fractal dimension may have a greater contribution to the difference in $LT_{50}$ between plants (Fig. 3).

### Physiological indicators

Compared with phase 1, the MDA content showed an increasing trend in phase 2, and there were significant ($p < 0.01$) differences among various rainfall events (Fig. 5). The content of MDA of $D_8$ increased by 25% from phase 1 to phase 2, which was less than that of $D_4$ (38%) and $D_2$ (59%). There were no major changes in MDA content between the three C at 25 °C/20 °C (day/night), but a significant difference appeared after the

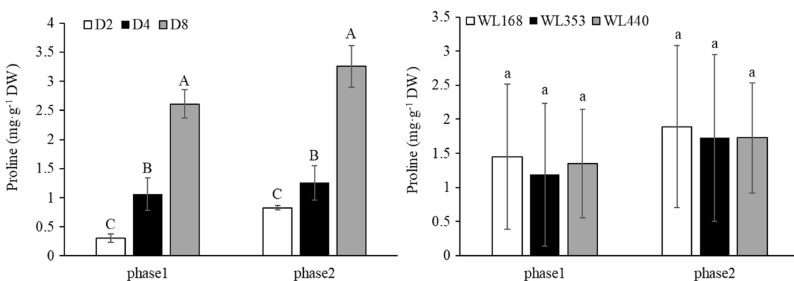

**Figure 6 Proline content under various simulated rainfall events for the three cultivars in the two phases.** Mean values ($n$ = 18) ± standard errors of the mean are shown. Different capital letters indicate a significant difference between three simulated rainfall events at the same phase at 0.01 level.

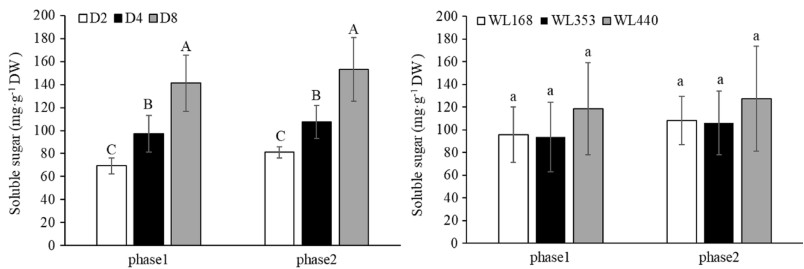

**Figure 7 Soluble sugar content under various simulated rainfall events and for the three cultivars.** Mean values ($n$ = 18) ± standard errors of the mean are shown. Different capital letters indicate a significant difference between the three rainfall events at the same phase at 0.01 level.

low-temperature experiment, and WL168 had a lower MDA content (100.9 nmol·g$^{-1}$ DW).

The Pro content of the three rainfall events showed significant ($p < 0.01$) differences in the two phases (Fig. 6), but there was no significant difference among the three C. $D_8$ had the highest Pro content at 2.61 and 3.26 mg·g$^{-1}$ DW in phase 1 and phase 2, respectively, followed by $D_4$ (1.06 and 1.26 mg·g$^{-1}$ DW) and $D_2$ (0.31 and 0.83 mg·g$^{-1}$ DW).

The soluble sugar content of the three treatments increased after cold acclimation (Fig. 7). In the same phase, SRE had a significant ($p < 0.01$) impact on the soluble sugar content of alfalfa root, which was manifested as an increasing trend as the rainfall interval increased. There were major effects ($p < 0.01$) on the soluble sugar content of the two phases among the three alfalfa C. WL440 had 118 and 127 mg·g$^{-1}$ DW soluble sugar content in phase 1 and phase 2, respectively, significantly higher than those of WL353 and WL168.

The starch content showed a downward trend at phase 2 for the three rainfall events in all three C (Fig. 8). In the two within-phase comparisons, SRE had a significant ($p < 0.01$) impact on the soluble sugar content of alfalfa roots, while the choice of C did not. The starch content of $D_8$ was 95.4 and 68.5 mg·g$^{-1}$ DW in phase 1 and phase 2, respectively, which was higher than in the other two treatments.

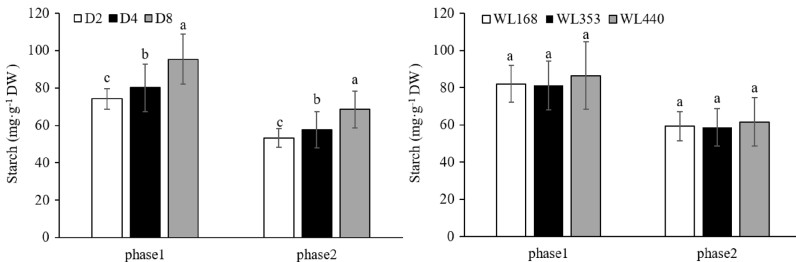

**Figure 8 Starch content under various simulated rainfall events for the three cultivars.** Mean values ($n = 18$) ± standard errors of the mean are shown. Different lowercase letters indicate a significant difference between the three rainfall events at the same phase at 0.05 level.

## DISCUSSION

### Root biomass

*Swemmer, Knapp & Snyman (2007)* and *Heisler-White, Knapp & Kelly (2008)* demonstrated that rainfall interval and rainfall amount are the key factors affecting the allocation of plant biomass. Rainfall events had no significant effect on the AGB of alfalfa, while longer rainfall intervals increased BGB and R/S (Table 1). This allometric growth relationship between root and shoot is an adaptive strategy by plants to various soil water conditions (*Padilla et al., 2009*). Research has shown that longer rainfall intervals can enhance the drought resistance of plants by promoting root growth and increasing energy distribution to underground parts (*Den Herder et al., 2010*; *Jangpromma et al., 2012*; *Padilla et al., 2009*). There was little difference in AGB between the three C, but the C with lower fall dormancy level had higher BGB and R/S. Root biomass has been proven to be an indicator of stress resistance (*Bloor, Zwicke & Picon-Cochard, 2018*), and the C with a lower level of fall dormancy distributes more energy to its roots. This also explains, to a certain extent, why alfalfa with a lower fall dormancy score has better cold resistance (*Oppelt, Kurth & Godbold, 2001*; *Xu et al., 2020b*).

### Root morphological traits and architecture

Root plasticity determines the ability of plants to survive in the ever-changing soil environment (*Tian, De Smet & Ding, 2014*). The primary root mainly plays a role of fixing, storing and transporting substances. It has a longer lifespan and slower metabolism, and there are no significant differences in response to various rainfall events (*Gruber et al., 2013*). Lateral roots are the main parts of the root system that absorb water and nutrient; they have a shorter lifespan, stronger metabolism and lower resistance to abiotic stress, and are more sensitive to changing environments (*Li et al., 2020*; *Padilla et al., 2009*; *Withington et al., 2006*). Through research on rice (*Oryza sativa* L.) and arabidopsis (*Arabidopsis thaliana* L.), *Pedersen et al. (2021)* found that a longer rainfall event results in an overall altered root system that ranges from changes in root system architecture, including fewer lateral roots and thicker primary root (*Hodge, 2004*; *Schwab et al., 1996*). Larger root surface area and specific root length are beneficial to the root water absorption efficiency system (*Hassouni et al., 2018*), but attention should be given to the cost of

water absorption under abiotic stress. Lateral roots are the most active part of the entire root system, and their faster turnover rate also increases the consumption of stored substances (*Guo, Mitchell & Hendricks, 2004*; *Rewald, Ephrath & Rachmilevitch, 2011*; *Withington et al., 2006*). $D_8$ has the shortest lateral root length (Table 3) and less consumption of stored substances, which can explain why alfalfa grown in intervals of longer rainfall showed better cold resistance in a subsequent low-temperature test (Fig. 2). Features such as a greater root volume and a small root surface area are evident after a longer rainfall interval, which enables plants to cope better with unpredictable soil conditions (*Pedersen et al., 2021*). Genotypes play an important role in the growth of plant roots under the same environment conditions and cultivation measures. Root systems of different cultivars respond differently to change in soil moisture (*Tron et al., 2015*). A field study was conducted to analyze the root system development ability of nine alfalfa cultivars; root biomass, primary root, lateral root, root surface area and root crown were significantly different among cultivars (*Zhang et al., 2002*). The three C presented various root trait responses to rainfall events: alfalfa with a low level of fall dormancy had shorter lateral roots and specific root length and a larger root volume, consistent with the results found by *Rimi et al. (2010)*. The spatial structure of a root system can further describe the distribution of roots in the soil. A study of the effects of rainfall on various species found that rainfall events have important effects on root system architecture, including in the roots of herbaceous plants (*Kume, Sekiya & Yano, 2006*). Root forks and average link length reflect the branching of plant roots. Root forks are related to water absorption efficiency (*Bauhus, Khanna & Menden, 2000*) and average link length represents the space expansion ability of the root system (*Walk, Van Erp & Lynch, 2004*). There is a negative correlation between root forks and average link length, and this depends on the soil conditions (*Schenk & Jackson, 2002a*; *2002b*). The results of rainfall events on alfalfa show that a root system under a longer rainfall interval has a longer average link length, while more root forks appear under shorter rainfall intervals (*Kong et al., 2014*). Root systems with a larger number of forks are more advantageous in resource-rich soil because they can quickly occupy the space available for rapid growth. Those with a longer average link length can improve competitiveness under water shortage because root overlap and unnecessary internal competition is reduced (*Bauhus, Khanna & Menden, 2000*; *Enquist & Niklas, 2002*; *Guswa, 2010*). We verified this trade-off relationship between root forks and average link length in the three C (Table 4). Topological index and fractal dimensions are both parameters that describe the root system architecture. Low-frequency heavy precipitation events are conducive to the development of herringbone branching in a root system (Table 4). Researcher has divided the branching patterns into dichotomous and herringbone according to the two extreme values of the TI, although the branching pattern of most plants falls between the two types (*Fitter, 1986*). A root system tends to dichotomous branching when the plant is in high-nutrient soil and the TI is close to 0.5; the branching pattern tends to herringbone when resources are scarce and the TI is close to 1 (*Fitter & Stickland, 1992*; *Glimskär, 2000*; *Li et al., 2020*; *Lynch, 2019*). Fractal dimensions are also important parameters in explaining the spatial structure of the root system. The change trend in fractal dimensions
and root forks is consistent in describing the use of space by roots, and our results agreed with those of *Li et al. (2020)* and *Dannowski & Block (2005)*. *Tron et al. (2015)* modeled the transpiration of 48 root architecture types under 16 drought scenarios and different soil structures and textures; they reported that root architecture did not fully explain plant water use and suggested relating specific root architecture with genotype and other characteristics. Genotypes differ in their localization of root biomass at different depths under water stress condition; tolerant genotypes produce deeper and more vigorous roots in the search for water (*Farooq et al., 2019*). *Manschadi et al. (2006)* studied root architectural traits in the adaptation of wheat to water-limited conditions using a drought-tolerant and drought-susceptible genotype; the tolerant genotype developed a compact vertical root system allowing it to extract less water during early growth stages but more as the crop matured. WL168 has the smallest root forks and average link length, and this is consistent with the study of *Farooq et al. (2019)*.

## Semi-lethal temperature and root traits

Alfalfa has to undergo a low-temperature and short-photoperiod process to maximize its cold resistance (*Theocharis, Clement & Barka, 2012*), and this characteristic is shown in all three cultivars (*Trischuk et al., 2014*). Whether at normal temperature or undergoing low-temperature stress, the semi-lethal temperature of $D_8$ was significantly lower than that of the other two (Fig. 2). Simultaneously, the cross acclimation of drought and low temperature has been confirmed in creeping bentgrass (*Agrostis stolonifera* L.) (*Zhang et al., 2015*) and alfalfa (*Xu et al., 2020b*), which provides theoretical support for improving cold resistance of alfalfa by water management. To further understand the relationship between water and cold resistance, we conducted a PCA analysis among root system traits and semi-lethal temperature (Fig. 4). The positive correlation factors that affect cold resistance of alfalfa mainly include below-ground biomass, crown diameter, root volume and topological index, while the increase in lateral root length, root surface area, root forks and fractal dimensions reduced the semi-lethal temperature. Some of our conclusions contradicted previous research (*Johnson et al., 1996*; *Larson & Smith, 1963*; *Liu et al., 2015*; *Smith, 1951*), but these differences were mainly caused by environmental factors rather than by low-temperature stress. The root crown is the most sensitive part of the root system to temperature changes and is crucial to overwintering and regeneration (*Bélanger et al., 2006*; *Janska et al., 2010*). The plasticity of the root crown is an important strategy for alfalfa in its adaptation to the cold climate in northern regions. As the crown diameter increases, the cold resistance of alfalfa is gradually enhanced (*Liu et al., 2015*; *Schwab et al., 1996*). The root biomass is related to the accumulation of organic matter and the herringbone branching is conducive to alfalfa's absorption of deeper water in cold winter, which contribute to improving the plant's cold resistance (*Larson & Smith, 1963*; *Viands, 1988*). Longer lateral roots accelerate nutrient consumption and are more susceptible to freezing under low-temperature stress (*Schwab et al., 1996*; *Withington et al., 2006*). Studies have confirmed that low-temperature stress inhibits plant growth (*Liu et al., 2019*; *Venzhik et al., 2011*), but little is known about how these characteristics affect cold resistance. The above analysis demonstrates the regulatory effect of rainfall events on

root system traits, and the next step is to manipulate these traits to enhance plant stress tolerance. It is possible that gene editing technology may allow plant root traits to be changed and so permit better adaptation to low-temperature environments (*Nakamichi et al., 2016*).

**Root physiological indexes**

In winter, physiological regulation such as osmotic regulation, antioxidant regulation and induction of antifreeze gene expression is the most important way for plants to adapt to low-temperature stress (*Bertrand et al., 2017*; *Choudhury et al., 2017*; *Theocharis, Clement & Barka, 2012*). Research has concluded that it is a significant correlation between cold resistance and the cell membrane, and plants with stronger cold resistance have a lower phase-transition temperature (*Anower et al., 2016*). MDA is the final product of cell membrane lipid redox, which can destroy the structure and function of proteins, nucleic acids and polysaccharides. Plants produce a larger amount of MDA under low-temperature stress, and its content has a significant negative correlation with the freezing tolerance of alfalfa (*Choudhury et al., 2017*; *Xu et al., 2020a*). In addition, excessive reactive oxygen species (ROS) can also affect cell activity. To reduce ROS damage, cells maintain their integrity through osmotic regulation (mostly of Pro, soluble sugar and starch) (*Anower et al., 2016*; *Bertrand et al., 2017*; *Castonguay et al., 2011*). Pro can act not only as an osmotic regulator or ROS scavenger, but also as a molecular chaperone to prevent cells from being damaged by low temperature (*Castonguay et al., 2011*; *Kishor et al., 2005*; *Nakashima et al., 1998*). Studies have proven that soluble sugar and starch are closely related to the freezing tolerance of alfalfa (*Anower et al., 2016*; *Castonguay et al., 2006*). Soluble sugar can improve the survival rate of alfalfa in winter in two ways: (1) it can act as an osmotic substance and cryoprotectant to lower the cell freezing point (*Castonguay et al., 2013*); and (2) it is a signal molecule that initiates a series of cold response mechanisms (*Bertrand et al., 2017*). Starch can be hydrolyzed into soluble sugar to improve the cold tolerance of alfalfa (*Sengupta et al., 2015*). In phase 2, the Pro and soluble sugar content of $D_8$ was still higher than that of the other two cultivars, which indicated that cold acclimation would maximize the cold resistance of alfalfa, but also that a longer irrigation interval can accelerate this process (*Xu et al., 2020a*; *Zhang et al., 2015*).

## CONCLUSIONS

In our study, a low-frequency heavy precipitation significantly enhanced the cold resistance of alfalfa compared with a high-frequency light precipitation. The increase in root biomass and crown diameter, and the decrease of lateral root length and root surface area may have contributed to the difference in $LT_{50}$ among plants receiving the different treatments. A fewer root forks and a bigger topological index may reduce the redundant consumption of the root system, which is beneficial to improve the cold resistance of alfalfa in a low temperature environment. Alfalfa with high fall dormancy grade has more below-ground biomass and less lateral root, and these characteristics are conducive to rapid acquisition of cold resistance. Simultaneously, a longer irrigation interval facilitated the accumulation of proline and soluble sugar content.

These conclusions provide support for winter water management of alfalfa and selection of varieties in areas of high latitude.

### Funding
This research was supported by the National Natural Science Foundation of China (Nos. 31772671 and 32071880). The funders had no role in study design, data collection and analysis, decision to publish, or preparation of the manuscript.

### Grant Disclosures
The following grant information was disclosed by the authors:
National Natural Science Foundation of China: 31772671 and 32071880.

### Competing Interests
The authors declare that they have no competing interests.

### Author Contributions
- Zhensong Li conceived and designed the experiments, performed the experiments, analyzed the data, prepared figures and/or tables, authored or reviewed drafts of the paper, and approved the final draft.
- Liqiang Wan conceived and designed the experiments, authored or reviewed drafts of the paper, and approved the final draft.
- Shuo Li performed the experiments, prepared figures and/or tables, and approved the final draft.
- Xianglin Li conceived and designed the experiments, authored or reviewed drafts of the paper, and approved the final draft.
- Feng He analyzed the data, prepared figures and/or tables, and approved the final draft.
- Zongyong Tong analyzed the data, prepared figures and/or tables, and approved the final draft.

### Data Availability
The raw measurements and data are available in the Supplemental Files.

### Supplemental Information
Supplemental information for this article can be found online at http://dx.doi.org/10.7717/peerj.11962#supplemental-information.

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
