# Peer review of "Plastic response of Medicago sativa L. root system traits and cold resistance to simulated rainfall events"

_PeerJ, doi:10.7717/peerj.11962_

## Round 0.1 · original submission · Major Revisions

Dear Dr Wan,

There are important issues raised by the reviewers, which mainly concern clarity in the presentation of methodology and statistical analysis. I am confident that addressing these issues will greatly improve the manuscript.

Reviewer 1 ·

Basic reporting

no comment

Experimental design

no comment

Validity of the findings

no comment

Additional comments

no comment

Annotated reviews are not available for download in order to protect the identity of reviewers who chose to remain anonymous.

Reviewer 2 ·

Basic reporting

1.Basic reporting
 In my opinion taking into account that I myself am not a native speaker of English language, the linguistic quality of the manuscript (MS) meets the standards of the journal. There is only a small number of minor corrections that I have detected and pointed out for the authors to correct. Please find detailed comment below.
 Literature references are sufficient and cover the necessary background.
 The structure of the MS conforms to the journal’s format.
 Figures are relevant, however some improvements are necessary concerning figure resolution and legends for both figures and tables. Please find my specific comments below.
 Raw data of all traits examined have been made available, thank you. There is a minor spelling issue i.e. column title of the starch values is written as strach. I also urge the authors to add titles in the first 3 columns and include the units of measurement for each trait in order to facilitate future readers.
 The article is self-contained and the results could become (extensive improvements are needed) relevant to the hypothesis.

Experimental design

2.Experimental design
 The work is within the Aims and Scope of PeerJ.
 The working hypothesis is clear and meaningful. The contribution that the MS is intended to have on filling knowledge gaps is clearly explained.
 The investigation performed is moderately rigorous (please see specific comments below). In general, technical and ethical standards are met. Statistical approach is the main issue.
 Methods description could be improved. Please find the relevant comments below.

Validity of the findings

3.Validity of the Findings
Although the technical part of the work seems overall nicely done, there is a major issue with the statistical analysis and therefore the presented results and consequently their validity. Authors should incorporate at least two more statistical tests and reconsider their whole statistical analysis in order to use a method that is more appropriate for their experimental design. Authors should also include in the description of their Statistical Analyses section (Materials and Methods), all the appropriate information regarding the PC analysis, that is now missing. Please find more details in the comments below.

Additional comments

4.General comments
In this MS, the authors aspire to provide further information on the role of mainly root morphological traits and their plasticity in cold tolerance of alfalfa. To do so, they investigated intraspecific variation and plasticity of root morphology and physiology among 3 cultivars by submitting them to 3 simulated rainfall treatments and examining their effects on selected root morphological and physiological indexes. They then tried to assess if these modifications result to an increase in cold tolerance of the cultivars using semi-lethal temperature as an indicator. The MS is of good quality and in general well-written (linguistically speaking), and could prove quite interesting for the readers of PeerJ. However, authors should consider making extensive alterations and additions for the improvement of some of the quite weak (at the moment) parts of the MS. Please find my specific comments per section below.

Title: Perhaps authors should consider using “plastic responses” instead of response, which is less informative.
The term cold resistance and tolerance seem to be used throughout the text interchangeably. Given the overall confusion around these two terms I would greatly appreciate it if authors could add a sentence for clarification purposes of their use of the terms. If resistance is used for describing the lack of adverse effects on the plant despite the existence of the stress factor, then cold tolerance is a better term for the title.

Abstract
Minor comments:
Line 19-22: I am not sure that the word phase clearly describes what the authors did. Of course there were two phases regarding the sequence of the work done, but if I understood correctly, actually these were not two phases but two different treatments: cold-treated plants and non-cold-treated plants. I would suggest a rephrasing that clarifies that from early on.
Line 31: perhaps examining instead of analyzing

Introduction
This section is well written, provides all the necessary background information and clearly states the goals of the article.
Minor comments:
Line 83: Given the logic of the plant x environment interactions, this sentence should be rephrased. Water management could prove important for increasing cold tolerance and not the other way around as implied.

Materials & Methods
This section needs some improvement for the better clarification of the experimental processes and data analysis.
Experiment Designs and treatments
Lines 107-116. It is not very clear which was the WHC throughout the experimental period for each of the SRE. What WHC did the D8 plants reach on the 8th day before the irrigation? In D4 plants, 4 days after the initial irrigation WHC reached 40%? Rewatering was done till WHC 60-65% for both D4 and D8 plants? Please explain this procedure in a more helpful for the reader way especially taking into account that rehydration affects the branching pattern.
Line 116: phase 1, please see comments above (abstract). Sampling was done on half of the total experimental plants i.e. 3x3x6? Please clarify.
Line 125: Do the authors mean the remainder plants i.e. the rest half of the experimental plants? Please clarify.
Authors should also explain why they chose to measure the non-cold-acclimated plants 3 days before the cold-acclimated plants, which means that there is a small but significant difference of the root system development that should have been taken into account for reasons of comparability. Especially considering that root structural modifications are quite fast and can become obvious after about 1 day, a phenomenon that the authors exploited for investigating the plasticity.
Authors should also make an effort to justify the choice of the timing (i.e. why 3 days of acclimation?) of the second sampling. How was the WHC change taken into account? Please justify.
Line 129: I understand the logic behind the light intensity reduction but this change adds another environmental factor that may affect the results. Has this been taken into account?

Measurements
Line 132: How much was WHC when plants were moved to the incubator? Were they rewatered before moving them? Were they rewatered during cold-accclimation?
Line 138: It would be very helpful if authors provided some visual material e.g a photograph of the root system.
Line 157-159: Please add a phrase explaining why the authors chose different temperatures for the two treatments.
Line 174: Please rephrase this sentence because it sounds like the material used for the measurement of LT was grounded and then used for the biochemical measurements which I guess and strongly hope is not the case.
Line 192: The incubation time is quite lower than the one proposed in Bates, 1973. If authors have modified the protocol, please mention and explain.

Statistical analyses
Line 225-227: Please clarify if the criteria for conducting a parametric test i.e. normality and homogeneity of variance were met. The tests done should also be mentioned. This would also justify the use of Pearson correlations instead of Spearman.
Line 229: According to the experimental design followed, a simple effect analysis is not the appropriate statistical test for comparing the variance. There are at least 3 factors i.e cultivar (3 levels), irrigation treatment (3 levels) and cold acclimation treatment or not (2 levels). If taking account temperature difference between the last treatments it would add 1 more factor with 2 levels. I suggest the authors should conduct a 3- way ANNOVA, and try to explain as suggested above (comments for Line 125) why temperature was not taken into account as an extra factor of possible variance.
Given that the authors have also conducted another statistical analysis i.e. PCA, the relevant information should be added at this section. Does the PCA include both the data from cold acclimated and non acclimated plants? Because the points in the biplot are only about 54 where it should be 18 treatments x 6 replicates =108 as mentioned in Line 112-113.
However, given that PCA is very useful as an ordination method I think that the authors should try exploit the very interesting information their analysis is providing them, instead of using it only as a correlation method. If so, a pairwise correlation matrix for all traits would be more suitable and just enough.
According to the PCA biplot, the grouping of the water treatments along the first axis provides a clearer picture of the interactions and the trade-offs among the examined traits and reinforces the results of the paper. Furthermore, a very useful and enlightening addition in the biplot would be to make a distinction among cultivars (eg by different shapes of the markers) and phase 1 and 2 data (eg by different shape outlines or even by circling them) and see how the cultivars, SREs and “phases” affect the ordination along the two axes. This would help explaining what each component represents in a more meaningful way. Perhaps authors should consider taking advantage of the different groupings/ordination on the biplot and try explain them in a more constructive way.

Results
Given that the statistical analysis is not appropriate for this kind of set of data, the presentation of the results is also not appropriate for understanding the intraspecific variation, the SRE effect, the cold effect and their interactions effect. The whole section needs reconstruction after the appropriate statistical analysis is done.
It would be more meaningful to compare the plasticity of each trait induced by the 3 SREs among the 3 cultivars and then check for probable links of such differences (either due to cultivar or SRE) with the cold acclimation response.
The comparison of the mean trait value among D2, D4 and D8 irrespective of cultivar and ”phase”, or among cultivars irrespective of SRE and ”phase” as presented in Tables 1,2,3,4, Figure 1,2, 3, 4, 5, 6 is not very helpful. Why compare the mean for each SRE/cultivar without taking into account the possible effects of the rest factors that the experiment includes? Authors need to follow a different statistical approach that is suitable for their experimental design.
Under each of the above mentioned tables and figure, the sample size is written as n=6. If I understood correctly, the mean for e.g. D2 derives from: either the mean of the mean values of the 3 cultivars therefore n=3, or the raw data i.e n=6 replicates x3 cultivars =18. Is that true? The same then goes for the rest…Please check that sample size is correctly written.
Another possible improvement concerns the table labels. It is not very helpful for the reader to see all these abbreviations in the label. I would suggest an approach like e.g.: Comparison of the biomass / morphological responses/ architecture/ physiological indexes among the three cultivars, SREs etc., namely a category/ grouping of the traits instead of abbreviations, that are correctly referred under the table.
Effect analysis should include all traits and treatments. So Table 2 is not very representative at the moment.
Table 5 is supposed to help elucidate which factor had the main effect? I do not get the point of this Table that it is not properly discussed, too. Please present these results appropriately and comment on them in a meaningful way.
And why are only morphological traits used and not physiological indexes? This is the case for the PCA, too. Why were only these included when it is expected that there are interactions among all traits, interactions that need to be investigated? Please justify the trait exclusion or adjust your analysis.
Even if Table 5 is kept, some more information is needed i.e. the sample size n=? is it the r or r2 presented. I am also wondering what kind of data SRE refers to. There is no numerical trait referred as so. Please explain.
Line 3: represent statistically significant
Figures seem to have a low resolution but perhaps it is a matter of the pdf file. Maybe authors should check that their figures format complies with the journals instructions.

Discussion
Overall the quality and structure of this section are the MS best attributes. However, this is not the case for the scientific arguments that need confirmation. Considering the extensive changes that need to be done for a more suitable statistical analysis and the consequences this will have on the results, the authors will probably have to confirm their results before discussing them. A good example of the problem that the present statistical approach creates is the results and therefore the discussion for AGB, RGB and R/S.
Given that Table 1 does not give any information about the comparison of the traits of the same cultivar at the different SREs (which is the approach followed for all traits), no safe conclusion can be extracted regarding the changes of AGB, BGB, R/S because the plasticity within cultivar under the different SREs has not been examined. It may prove that there is no difference at all but at the moment there is no evidence that can support that. Also one cultivar may be more plastic than the other ones and this may eliminate the variation when not addressing the plasticity of each cultivar separately. And unfortunately this is the basic weakness of all results discussed.
I strongly suggest that after the reconsideration of the statistical approach, authors need to confirm the validity of their arguments and conclusions.

Reviewer 3 ·

Basic reporting

This manuscript present results on the impact of different rainfall intervals on alfalfa freezing tolerance in relation with root traits and the concentration of stress-regulated metabolites under controlled conditions. The authors found that longer irrigation interval was beneficial in improving the cold resistance of alfalfa. While this paper adds new knowledge in the field, particularly on the impact of rainfall patterns on root traits, major improvements are needed to the manuscript to better explain what was done.

Abstract: The authors use freezing tolerance and cold resistance to express similar concepts. They should explain the difference between the terms and choose the one that relates to the concept they want to express throughout the manuscript.
Same observation for ability to survive winter and/or winter survival: these parameters were not measured in this study and winter survival depends on many factors, not only freezing tolerance.

Experimental design

My comments are regrouped in the section on General comments to the author

Validity of the findings

My comments are regrouped in the section on General comments to the author

Additional comments

This manuscript present results on the impact of different rainfall intervals on alfalfa freezing tolerance in relation with root traits and the concentration of stress-regulated metabolites under controlled conditions. The authors found that longer irrigation interval was beneficial in improving the cold resistance of alfalfa. While this paper adds new knowledge in the field, particularly on the impact of rainfall patterns on root traits, major improvements are needed to the manuscript to better explain what was done.
Abstract: The authors use freezing tolerance and cold resistance to express similar concepts. They should explain the difference between the terms and choose the one that relates to the concept they want to express throughout the manuscript.
Same observation for ability to survive winter and/or winter survival: these parameters were not measured in this study and winter survival depends on many factors, not only freezing tolerance.
Abstract L. 30. Contrary to the abstract, Fig. 6 shows higher concentration of starch in D8 than in D2 and D4. Rewrite.
L42 : Replace winter failure by crop failure due to winter conditions
L49: Change ‘There are two main impacts of climate change on plant overwintering: precipitation and temperature’ by ‘Two major factors linked to climate change are likely to affect plant winter survival: changes in precipitation and temperatures (Bélanger, 2002; CCAF, 2001).
L.59: Water management plays an important role in the winter hardiness of alfalfa because freezing injury is mainly caused by cell dehydration: The logic behind this sentence is not true: freezing injury is caused by cell dehydration because water freezes in extracellular space, which is not directly linked to water management. Rewrite.
L. 57 and L.58 The authors use freezing tolerance and cold resistance to express similar concepts. They should explain the difference between the two terms and choose the one that relates to the concept they want to express throughout the manuscript.
L.66 …were found to have a greater impact on herbaceous plants. As compared to what? Please complete the sentence.
L. 76-78: There is a very brief introduction on malondialdehyde (MDA), proline, soluble sugars and starch related to their antioxidant capacity and osmotic regulation in plants. The authors state that these compounds are closely related to cold resistance of alfalfa. It should be specified if the relationship is positive or negative and the role of each compound should be briefly explained in relation to freezing tolerance in this section.
L.80 and 82. On cold resistance or freezing tolerance? It is not the same thing. What did you measure in your experiment?
L.96: It is stated that a ratio of 4:1 sandy soil and nutrient soil mixture was used. What is a nutrient soil mixture? This needs clarification.
L114-131. This part is not easy to follow. If I understand, the plants were three weeks old when the treatments started. Plants received the different irrigation treatments during 8 weeks, followed by a first sampling. Then the remaining plants were cold acclimated 72 hours followed by a second sampling. A table or Figure explaining clearly the treatments and sampling should be provided.
L 131. If the acclimation treatment lasted 72 hours, how could the plants be watered as in the previous phase? Does it means that only D2 plants were watered during this period? It should be clarified.
L. 152-153. To assess electrolyte leakage, the authors sampled the crowns that were sliced into 9 pieces about 2-3 mm and put into 2-mL centrifuge tubes. It should be explained how were divided the crowns into 9 pieces of 2-3 mm since Table 3 reports crown diameters of 3 to 4 mm. Were several plants pooled together? Furthermore, I am not confident that actual plant damages could be assessed by using such a small quantity of materials (2mm). Did you validate your method with other plant survival measurements? Crowns are very small organs at the intersection of roots and stems so please explain how it was sampled.
L.173. Since the roots were used for the analysis of morphological traits, it is not clear on which plant material the root physiological indexes were measured. I understand that 0.2 g were used for MDA, 0.2 g for proline, and 0.2 g for soluble sugars, and a part was used for LT50 while the dry weight of the belowground biomass was around 0.6 and 0.9 g (Table 1). Were several plants pooled together to perform all these analysis? This part needs clarification. How big were the plants?
L. 174. After measuring electrical conductivity the remaining samples were ground into powder and freeze-dried to determine their SS: is it root samples or crown samples ( it is under the subsection on root physiological index). This should be clarified since crowns and roots biochemistry highly differ.
To avoid the degradation of the samples, it is recommended to freeze-dry the samples before grinding. Is it what was done? Please specify.
L.236. How was measured the biomass? It is not in the M&M section. The following abbreviation BGB, AGB, C, R and S should be spelled out on their first appearance in the manuscript.
L.272 and Fig. 1. How could you determine that damages were observed in alfalfa exposed for 1.5 h at 4C. The damages could be due to the protocol described since tubes were placed at 4°C for 2 h and then transferred into ice for 2 h before their transfer in a ZX-5C for the freezing test at temperature varying from 8, 6, 4, 2, 0, −2, −4, −6. The damages could be due to the two hours into ice. This should be clarified since alfalfa is usually not damaged by a short exposure at 4C.
L.293. What do you mean by growth rate (The MDA growth rate)? The increase from phase 1 to phase 2? It is not a growth rate.
The discussion is good.

---

## Round 0.2 · Minor Revisions

All the line numbers below refer to the “R1_with_ tracked_changes.docx” file

General remarks

Resistance vs. tolerance: while your answer to Reviewer#2’s comment about the title is that you are going to use cold tolerance, you finally use cold resistance. At the same time your answers to Reviewer#3 is that you measure cold resistance. Please check which title and which term is more appropriate for your work and make the relevant modifications.

There are several cases of not incorporating text or image in the R1, although you definitely stated in your rebuttal letter that you have entered it in the revised MS. This also holds for deletions or modifications asked by the reviewers and accepted by you.

Please carefully address all the below-mentioned comments and corrections.

Specific comments
L93, replace “for” with “since” to make sense.

L108: please provide the company name instead of the name of the country of origin (Lithouania).

L136: please use mL for consistency with the other parts of the MS.

L152: you should add the information about watering of the plants in the acclimation phase given in your response to Reviewer#3 relevant comment:
[L 131. If the acclimation treatment lasted 72 hours, how could the plants be watered as in the previous phase? Does it means that only D2 plants were watered during this period? It should be clarified.
We apologize for this omission. As shown in Figure 3, the plants were subjected to different irrigation treatments for a total of 69 days. In order to keep the total irrigation amount of all treatments consistent, we performed the last irrigation on the 65th day (the day before the temperature was 5/0℃). No watering was performed in all treatments after the temperature was 5/0℃.]

L158-159, in your response to Reviewer#2 suggestion for an image, you gave a figure: “Figure 2 Scanning image of root system under three simulated rainfall events”. But you have not incorporated it in the R1. Please do it and define it as Image 1 in order to keep the original numbering of your figures.

L172-173. Concerns raised by Reviewer#3 were not answered in the MS. So, I suggest that you incorporate your explanation and the citation as written in your Rebuttal letter.

L174. You have NOT deleted the ice treatment, as you mentioned in your relevant response to Reviwer#3.

L195-198. Please check the sentence and change it to make sense.

L194-248. It is not clear that the root physiological Indices refer to crown. Though you mention it in L139, I suggest adding it also here.
Also in this section: Please clarify your fresh sample handling. In your relevant answer to Reviewer#3 you mention that you used liquid nitrogen for instantly freezing the fresh material. Nevertheless, in the following response you mention something different. Which was exactly the procedure followed. Please add it clearly to the MS.
[To avoid the degradation of the samples, it is recommended to freeze-dry the samples before
grinding. Is it what was done? Please specify.
We apologize for this omission. Freeze-dried sample was not a correct term here. We kept 0.2 g fresh sample at -80℃ and ground it when measuring physiological indexes (we have clarified in the manuscript).]

L313-315, rephrase the sentence to make sense and correct that different colors denote different SRE and shapes refer to cultivars.

Statistics section. Please re-write the whole section and specifically (and clearly) indicate the statistical tests you used for which variables and effects.
In your response to Reviewer#2 comments you argue that:
“We conducted Kolmogorov-Smirnov test and Levene test, and the data in this experiment obeyed a normal distribution and satisfied the homogeneity of variance. LSD was used to conducted analysis of variance”.
Why did you use a non-parametric test (Kolmogorov-Smirnov) if both normal distribution and homogeneity of variance were satisfied?
LSD follows ANOVA for multiple comparisons, it is not analysis of variance by itself.
You do not mention two-way ANOVA in you MS.
LSD in L253 is wrong.
Please re-phrase L259 sentence to make sense.

Discussion
A text discussing the between-cultivar differences and in which measured parameter may be ascribed will benefit this section a lot. It was emphasized by 2 of 3 reviewers but was not adequately addressed in Discussion of R1.

Figure 1 - caption: there must be an explanation of the two given temperatures (day/night).
Figure 3 – caption: please i) indicate that different shapes denote cultivars and colors SREs, ii) give once again the full term of each of the 10 variables depicted (e.g. RSA, root surface area; RF, root forks; etc.).

---

## Round 0.3 · accepted · Accept

All the issues were addressed.